# In Vivo Anticancer Evaluation of 6b, a Non-Covalent Imidazo[1,2-*a*]quinoxaline-Based Epidermal Growth Factor Receptor Inhibitor against Human Xenograft Tumor in Nude Mice

**DOI:** 10.3390/molecules27175540

**Published:** 2022-08-28

**Authors:** Zahid Rafiq Bhat, Manvendra Kumar, Nisha Sharma, Umesh Prasad Yadav, Tashvinder Singh, Gaurav Joshi, Brahmam Pujala, Mohd Raja, Joydeep Chatterjee, Kulbhushan Tikoo, Sandeep Singh, Raj Kumar

**Affiliations:** 1Department of Pharmacology and Toxicology, National Institute of Pharmaceutical Education and Research, Sahibzada Ajit Singh Nagar 160062, India; 2Laboratory for Drug Design and Synthesis, Department of Pharmaceutical Sciences and Natural Products, School of Health Sciences, Central University of Punjab, Bathinda 151401, India; 3Laboratory of Molecular Medicine, Department of Human Genetics and Molecular Medicine, Central University of Punjab, Bathinda 151401, India; 4Integral BioSciences Pvt. Ltd., C-64, Hosiery Complex, Phase-II, Noida 201306, India

**Keywords:** EGFR inhibitor, lung cancer, xenograft mice model, imidazo[1,2-*a*]quinoxaline, in vivo, microsomal stability, immunoblotting

## Abstract

Tyrosine kinase inhibitors are validated therapeutic agents against EGFR-mutated non-small cell lung cancer (NSCLC). However, the associated critical side effects of these agents are inevitable, demanding more specific and efficient targeting agents. Recently, we have developed and reported a non-covalent imidazo[1,2-*a*]quinoxaline-based EGFR inhibitor (**6b**), which showed promising inhibitory activity against the gefitinib-resistant H1975(L858R/T790M) lung cancer cell line. In the present study, we further explored the **6b** compound in vivo by employing the A549-induced xenograft model in nude mice. The results indicate that the administration of the **6b** compound significantly abolished the growth of the tumor in the A549 xenograft nude mice. Whereas the control mice bearing tumors displayed a declining trend in the survival curve, treatment with the **6b** compound improved the survival profile of mice. Moreover, the histological examination showed the cancer cell cytotoxicity of the **6b** compound was characterized by cytoplasmic destruction observed in the stained section of the tumor tissues of treated mice. The immunoblotting and qPCR results further signified that **6b** inhibited EGFR in tissue samples and consequently altered the downstream pathways mediated by EGFR, leading to a reduction in cancer growth. Therefore, the in vivo findings were in corroboration with the in vitro results, suggesting that **6b** possessed potential anticancer activity against EGFR-dependent lung cancer. **6b** also exhibited good stability in human and mouse liver microsomes.

## 1. Introduction

Lung cancer is common in males and females with a very low survival rate as it has a poor prognosis. Two major types of lung cancer are prevalent: small-cell lung carcinoma (SCLC) and non-small-cell lung carcinoma (NSCLC), with an occurrence rate of 15% and 85%, respectively. NSCLC can be further sub-classified as squamous-cell carcinoma, adenocarcinoma, and large-cell carcinoma. Adenocarcinoma is one of the prevalent types of lung cancer, irrespective of age (40% of all lung cancers), arising from small airway epithelial, type II alveolar cells from where mucus secretes [1,2]. The major risk factors causing NSCLC are environmental factors including cigarette smoking and genetic risk factors. The treatment of lung cancer can range from surgery to radiation to chemotherapy and targeted therapy [3,4,5].

Current treatment options include (a) surgery for patients who have stage I, II, and IIIA lung cancer; (b) radiotherapy, which leads to restriction or eradication of tumors from specific sites in the body, and (c) chemotherapy, for about 40% of recently diagnosed lung cancers patients are stage IV. For stage IV NSCLC, cytotoxic chemotherapy is the first-line therapy and includes platinum (cisplatin or carboplatin) plus etoposide, paclitaxel, gemcitabine, docetaxel, vinorelbine, irinotecan, or pemetrexed [6].

The epidermal growth factor receptor (EGFR) contains an intracellular tyrosine kinase domain with an extracellular epidermal-growth-factor-binding domain [7,8]. On binding to its ligand, the EGFR is auto-phosphorylated by intrinsic tyrosine/kinase activity, causing activation of several signal transduction cascades resulting in more aggressive tumor phenotypes [9,10,11]. In the case of NSCLC, EGFR mutations are emerging, leading to more resistant cases against the existing chemotherapeutic agents. Several mutations in the EGFR have been linked to cases of NSCLC [12,13]. These EGFR-positive NSCLC cases have been reported to be resistant to standard chemotherapeutic drugs [14]. Advances in cancer research have identified efficient tyrosine kinase inhibitors that, by interfering with EGFR function, have shown to be beneficial in EGFR-dependent cancers cases [15]. Moreover, the resistance associated with erlotinib, gefitinib, etc. (Figure 1A), owing to T790M and C797S mutations in the EGFR and the severe side effects of these agents are inevitably demanding more specific and efficient targeting agents [16,17,18,19]. Recently, we reported novelly designed, synthesized non-covalent imidazo[1,2-*a*] quinoxaline-based EGFR inhibitors. Out of a series of 30 inhibitors, the **6b** compound emerged as the most promising EGFR inhibitor with an IC_50_ value of 211.22 nM when tested for EGFR inhibitor potential [20,21,22]. Furthermore, **6b** exhibited a great anti-proliferative effect against gefitinib-resistant H1975 lung cancer cells with mutant EGFR (L858R/T790M) expression (Figure 1B), gefitinib-resistant H1975 cells (IC_50_ = 3.65 µM) as compared with standard gefitinib (IC_50_ > 20 µM). We, therefore, aim to investigate the anticancer activity of **6b** (Figure 1A) in a xenograft nude mice model.

## 2. Results

### 2.1. ***6b*** Portrayed Profound Anticancer Potential in Comparison with Standard Drug Gefitinib in an A549-Cell-Induced Lung Cancer Xenograft Model

We found a promising anticancer activity of **6b**, especially at a high dose compared with gefitinib as standard in A549 cells (in vitro). Further, evaluation of the antitumor efficacy of **6b** in vivo requires the development of a xenograft animal model for lung cancer. To ensure whether the **6b** compound can also translate the anticancer effect in vivo, we developed a xenograft model of lung cancer by subcutaneous injection of A549 cells into the flank of nude mice and evaluated its efficacy. Tumors were allowed to grow to around 80 mm^3^ tumor volume; after that, the animals were divided into four groups (n = 7). The tumor volume was checked twice weekly for up to three weeks. Gefitinib 30 mg/kg (GEF) was used as a positive control in our study. GEF alone caused suppression of tumor volume to some extent. High-dose (HD) **6b** (30 mg/kg) significantly decreased the size of the tumors when compared with low dose (LD = 10 mg/kg) and the tumor control group. **6b**-HD exerted the maximum effect on the tumor regression, as evident from Figure 2. Percentage change in body weight was assessed in all animals of each group to determine the toxicological effects of each treatment. However, the % change in body weight was significant in **6b**-HD and the gefitinib treatment group (Figure 2C). In addition, the Kaplan–Meier survival analysis indicated that **6b**-HD and gefitinib improved the survival of the mice (Figure 2D).

Representative macroscopic images of surgically removed tumors are shown in Figure 3A. The percentage tumor growth inhibition was highest, and average tumor weights were lower in the **6b**-HD (30 mg/kg) and gefitinib groups than in the other groups (Figure 3B). The % tumor growth inhibition in **6b**-LD (10 mg/kg), **6b**-HD, and gefitinib groups were 16%, 33%, and 27%, respectively. Moreover, the respective treatment groups showed cytotoxic effects on the tumor, as evident from the H&E staining (Figure 3C), where both **6b** and gefitinib showed marked cytotoxic damage.

In lung adenocarcinoma, activated EGFR-mediated signaling causes the activation of many downstream signaling pathways that help in lung cancer cell proliferation, growth, and resistance to chemotherapies. In EGFR-activated cancer cells, the PI3K/AKT pathway is frequently altered, which aids cancer cell survival and resistance. TKIs reduce downstream signaling and inhibit cancer cell proliferation and survival by inactivating EGFR-mediated signaling [23,24,25]. Compared with tumor tissues isolated from untreated mice, the Western blot assay results show that compound **6b** decreased the AKT, P-AKT, and PI3K proteins in tumor masses isolated from mice treated with LD and HD. Compared with the tumor mass isolated from control untreated mice, the compound **6b** increased the level of PTEN protein in tumors isolated from mice treated with positive control gefitinib and both doses of **6b**. Compound **6b** decreased the PI3K/AKT pathway in tumor masses treated with it, according to the findings.

### 2.2. Discussion

In support of in vitro studies, in vivo antitumor efficacy of the **6b** compound using a xenograft model of NSCLC cancer induced by transplanting A549 cells into nude mice revealed that **6b** alone significantly suppressed tumor volume, especially at a high dose of 30 mg/kg. This result further supports our observations carried out in in vitro evaluation. Moreover, the average tumor weight was significantly low in all the treatment groups and lowest in the **6b** and gefitinib groups compared with the control group. In a similar line of observation, we found that the % change in body weight observed was minimal with the **6b**-HD and gefitinib groups. This could occur due to the extent of cytotoxic damage to the tumor by the respective drug treatments, as evident from H&E staining. The maximum level of cytotoxicity was found in tumors of **6b**-HD- and gefitinib-treated nude mice. Previously reported studies have represented that increase in the interstitial space leads to a high extent of penetration of paclitaxel and doxorubicin, which results in apoptosis in head and neck cancers and prostate cancer, respectively [26,27].

Similarly, we calculated the % tumor growth inhibition (% TGI) and found that it was significantly improved in all treatment groups in comparison with the tumor control group and was highest in **6b**-HD- and gefitinib-treated animals. Anticancer agents that target the tumor microenvironment and cancer cells have been reported to prolong survival and initiate a marked inhibition of tumor growth [28,29,30]. Interestingly, the Kaplan–Meier survival analysis revealed that **6b**-HD and gefitinib improved the survival of the mice markedly, followed by the survival status of mice treated with **6b**-LD and the control groups, respectively. However, more studies are required to further extrapolate the role and efficacy of **6b** compared with other chemotherapeutic agents and in other types of cancers.

Further, we performed the immunoblotting assay on the isolated tissue to elucidate the impact of in vivo anticancer potential on the expression of key proteins regulating the EGFR. We studied the implication of anticancer potential using PCNA, PTEN, AKT, *p*-AKT, and PI3K using both tissue samples of **6b** with low and high doses. The significant findings from the analysis (Figure 4A) revealed that PTEN expression was upregulated. The literature findings suggest that PTEN loss is correlated not only with the resistance to EGFR inhibitors in mutant strains but is also associated with late endocytic trafficking of EGFR and further activates Akt and EGFR [31,32]. As a consequence, PTEN was upregulated, and Akt protein expression was found to be downregulated. The Akt, a downstream ally of EGFR, is frequently expressed in breast and lung cancer; its upregulation is also associated with resistance to EGFR inhibitors, EGFR recycling, and cancer cell survival migration [31,32]. The expression of PI3K was significantly reduced in tissue samples treated with a high dose of **6b**. This is important since PI3K expression regulates cancer cell survival, proliferation, and resistance to chemotherapy. Further, the level of PI3K is found to be significantly upregulated in numerous solid cancers. Moreover, the EGFR induces PI3K-mediated Akt activation [33,34].

Altered gene expression in lung cancer cells is linked to a reduction in tumor mass and tumor cell resistance after treatment with TKIs. In a mouse lung tumor, we examined the effect of the compound on the expression of the genes Twist, Zeb1, and Hif-1α. Twist and Zeb1 genes help epithelial–mesenchymal transition (EMT) in cancer cells [35]. EMT increases the stemness of cancer cells and also causes metastasis. Cancer cells show resistance to cancer therapeutics by enhancing the expression of EMT regulatory proteins. The compound **6b** decreased the EMT-regulating genes Twist and Zeb 1 (Figure 4B,D) in mice tumors upon treatment compared with untreated mice lung carcinomas. Compound **6b** had a greater effect on Zeb1 than twist. Hif-1α helps to regulate angiogenesis, tumor growth, metastasis, and metabolic reprograming. The expression of the Hif-1α gene was decreased in gefitinib- as well as **6b**-treated lung tumor cells in a dose-dependent manner (Figure 4C). EGFR is associated with EMT and metastasis in many tumors, including lung cancer [36,37,38]. Mice lung cancer cells treated with compound **6b** inhibited the expression of the Twist and Zeb1 genes, which are responsible for EMT, leading to metastasis of tumor cells. EMT causes resistance against TKIs and a poor prognosis and diagnosis for lung cancer patients [39]. Targeting the EMT-promoting genes in lung cancer aids in lung tumor treatment and overcoming resistance against TKIs [35]. Compound **6b** inhibited angiogenesis, tumor progression, metabolic reprograming, and metastasis by inhibiting EGFR by decreasing Hif-1α expression upon treatment. Previous in vitro, in vivo preclinical, and clinical studies have shown that many growth factors, including EGFR, activate Hif-1α. In clinical studies, NSCLC Hif-1α is associated with poor diagnosis and relapse [40,41,42,43].

Further, we performed the microsomal stability assay to understand the intrinsic clearance and in vivo stability of **6b** [44,45]. The stability assay was performed using human liver microsomes (HLM) and mouse liver microsomes (MLM), employing verapamil and carbamazepine as controls. During the study, we measured three parameters: % drug remaining at 30 min, half-life (t½), and intrinsic clearance (CLint). The analysis (Table 1) revealed compound **6b** exhibited good stability in human and mouse liver microsomes.

## 3. Conclusions

In short, we established the potential of **6b** as an anticancer agent and it was observed to improve the survival profile in mice. The immunoblotting analysis revealed that **6b** effectively modulates the downstream pathways mediated by EGFR. **6b** also exhibited good stability towards human and mouse liver microsomes. Further, it would be interesting to explore the potential of **6b** in the future, besides serving as a lead for structural modifications, in a gefitinib-resistant H1975-induced xenograft model to understand its impact in EGFR mutated cancers.

## 4. Experimental

### 4.1. Synthesis of 1-((3,4,5-Trimethoxybenzylidene)amino)-4-(3,4,5-trimethoxyphenyl)imidazo[1,2-a]quinoxaline-2-carbonitrile (***6b***))

Intermediate **4** was synthesized as per previously developed protocol in our laboratory [10,11], via diaminomaleonitrile (**1**) condensation with triethyl orthoformate in 1,4-dioxane to produce compound **2**, which undergoes a substitution reaction with 1,2-diaminobenzene to produce compound **3**. Compound **3** undergoes the ring formation, which yields compound **4** upon treatment with base (1 M KOH). The target compound, **6b**, was synthesized from compound **4**, which undergoes a Pictet–Spengler reaction when it reacts with commercially available 3,4,5-trimethoxybenzaldehyde (**5**) (Figure 1). The physical data of **6b** is given below:

**Yield**: 73%; **Color**: yellow; **mp**: 143–145 °C. **^1^H NMR** (Appendix A; 600 MHz, CDCl_3_, TMS = 0) δ: 9.02 (1H, s), 9.00 (1H, d, *J* = 3.9 Hz), 8.14 (1H, d, *J* = 6.6 Hz), 8.05 (2H, s), 7.64 (1H, t, *J* = 7.5 Hz), 7.60 (1H, t, *J* = 7.8 Hz), 7.32 (1H, s), 4.02 (6H, s), 4.01 (6H, s), 4.00 (3H, s), 3.95 (3H, s). **^13^C NMR** (Appendix A; 151 MHz, CDCl_3_, TMS = 0) δ: 164.27, 153.88, 153.10, 144.99, 143.36, 140.96, 137.01, 135.84, 130.43, 130.23, 130.16, 128.73, 127.67, 127.60, 117.94, 115.60, 107.51, 107.13, 107.03, 104.24, 61.25, 61.03, 56.46, 56.41.

### 4.2. Animal Study Design

The animal experiments were performed as per the guidelines of the Committee for the Purpose of Control and Supervision of Experiments on Animals (CPCSEA) with the approval of the Institutional Animal Ethics Committee (# IAEC/19/14). Four–five-week-old male athymic nude mice were procured from Vivo Bio Tech Ltd., Hyderabad, India. Animals were kept in an individually ventilated caging (IVC) system in a pathogen-free environment with standard conditions of temperature: 22 ± 1 °C, humidity: 50 ± 10%, and a 12 h light/dark cycle maintained in the National Toxicology Center (NTC) in the National Institute of Pharmaceutical Education and Research (NIPER), Mohali. Throughout the study, animals were fed with a standard rodent chow diet and water ad libitum.

Xenograft, the nude mice model, was developed by injecting A549 (1 × 10 ^6^ cells, suspended in Matrigel™) subcutaneously into the right flank of each mouse. Tumor-bearing mice were randomly divided into four groups with seven animals per group after attaining an anticipated tumor volume of >80 mm^3^. The four groups included tumor control (TC), low-dose **6b** (**6b**-LD  = 10 mg/kg), high-dose **6b** (**6b**-HD = 30 mg/kg), and gefitinib (GEF = 30 mg/kg) as a positive control group. Treatment was given intraperitoneally once a week for a period of 21 days. Tumor volume and body weight were measured weekly during the study. Vernier caliper was used to measure the size of A549-cell-induced tumors weekly, starting from two weeks after inoculation. Tumor size was calculated by using the formula: length/2 × width/2 × 3.14, and the average tumor size for individual treatment groups was determined.

### 4.3. Histopathological Analysis of Tumors

Tumors were surgically removed from animals, sliced, and fixed in 10% *v*/*v* formal saline. For the histological study, the sliced tumors were processed and embedded in paraffin; sections of 5 µm were mounted on poly-L-lysine-coated glass slides. Sections were then stained with hematoxylin & eosin (H&E) to observe the structural alterations as described. The coverslip was mounted using DPX, the sections were observed at 40× magnification using the OLYMPUS BX51 microscope, and the images were captured with OLYMPUS DP 72 camera attached to the microscope.

### 4.4. Western Blot Analysis

Tissue that had been kept at −80 °C was removed and placed on ice for 5 min. A total of 20 mg of tissue was diced and placed in a 1.5 mL microcentrifuge tube kept on ice. The tissue was lysed to extract the proteins using RIPA lysis buffer with a cocktail (IX) of phosphatase and protease inhibitors, followed by sonication for 30 min. After that, the mixture was centrifuged, and the supernatant was collected for protein estimation using the Bradford assay. Thereafter, immunoblotting was performed using SDS-PAGE for the separation and resolving of individual proteins by their molecular weights. The proteins were further transferred to nitrocellulose membranes, and blocking was performed in 3% powdered milk solution. Further, the membrane was washed using TBST buffer and was incubated in primary antibody (CST) for 3 h. After incubation, the membrane was washed in TBST and re-incubated for 1h in secondary antibody. The immune complex was detected using ChemiDoc^TM^ (BioRad) after washing the membrane with TBST [46].

### 4.5. In Vitro Microsomal Stability

Compounds were incubated at a final concentration of 1 µM with human (HLM) and mouse (MLM) at 0.5 mg/mL and NADPH (1 mM) in 100 mM phosphate buffer (pH 7.4) at 37 °C. The reactions were terminated at 0 and 30 min by the addition of cold acetonitrile containing propranolol (50 ng/mL) as an internal standard. The reaction mixtures were partitioned by centrifugation at 15,000 rpm for 15 min, and the resulting supernatants were analyzed for the remaining test article by LC-MS/MS (Shimadzu Nexera UPLC with an AB Sciex 4500 detector). All studies used Verapamil hydrochloride as a positive control [47].

## Data Availability

The data presented in this study are available on request from the corresponding author.

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
