# Peer review of "In Vivo Anticancer Evaluation of 6b, a Non-Covalent Imidazo[1,2-a]quinoxaline-Based Epidermal Growth Factor Receptor Inhibitor against Human Xenograft Tumor in Nude Mice"

_molecules, 2022, doi:10.3390/molecules27175540_

Round 1

Reviewer 1 Report

I am in receipt of the research article by Kumar et. al. on “In vivo anticancer evaluation of 6b, a non-covalent imidazo[1,2-a]quinoxaline-based epidermal growth factor receptor inhibitor against human xenograft tumor in nude mice”. Despite the availability of a variety of EGFR inhibitors, the discovery of new EGFR inhibitors combating resistance is still warranted.  Authors previously reported a series of imidazo[1,2-a]quinoxalines (including 6b compound) along with their EGFR inhibitory activity (Molecules 2021, 26, (5), 1490).  In the present work, the authors have disclosed their findings on in vivo assessment of 6b against human xenograft tumor in nude mice. The results of the study are quite impressive and supported by experiments such as histological examination featuring the cancer cell cytotoxicity ability of the 6b compound characterized by cytoplasmic destruction observed in the stained section of tumor tissues of treated mice. Further, the immunoblotting and qPCR results signified that 6b inhibited the EGFR in tissue samples and consequently altered the downstream pathways mediated by EGFR leading to a reduction in cancer growth. In addition, the microsomal stability of 6b in HLM and MLM, showing descent stability is another strength of the manuscript. The work is interesting and has been systematically carried out. Therefore, I strongly recommend the article for its publication in this prestigious Journal, Molecules, with minor corrections.

1. What is superscript 4 in section 4?

2. Please check the sentences for grammatical mistakes

Author Response

Response: Thank you very much for recommending our manuscript for publication.

  1. What is superscript 4 in section 4?

Response: That was a typo. We have removed this.

  1. Please check the sentences for grammatical mistakes

Response: We have language edited the manuscript.

Reviewer 2 Report

In this paper the authors determined the in vivo anticancer potential of compound 6b, previously identified by the research group. The compound inhibit wt-EGFR with comparable potency than gefitinib, and was supposed to overcome the T790M resistance. The in vivo study herein presented confirm the potential of 6b as anticancer compound. Although this result is of interest, the authors should better explain some unclear points before the paper could be considered for publication:

1. which is(are) the advantage(s) of 6b with respect to gefitinib that could justify a potential clinical application of the new compound?

2. in the introduction the authors reported a superior activity of 6b in gefitinib-resistant H1975 cells with respect to gefitinib. Nevertheless, they then used a xenograft model of A549 cells which are sensitive to gefitinib. Why? It sould have been better and more interesting to evaluate the in vivo potential of 6b in a gefitinib-resistant model. Clearly, I can not suggest to re-run the in vivo experiments on a different cancer model, but this point needs to be properly discussed and clarified in the text

3. how LD and HD have been selected? Besides, the LD (10mg/Kg) and HD (30 mg/Kg) are not that different. It would have been better to use a ten-fold difference in the two doses. Please clarify

Some minor points:

1. the English style is not fine, and several sentences need to be rephrased

2. check for type errors. Just as an example, in Figure 4B-D there is a mistake (Zefitinib in place of Gefitinib). Also, the figure is of very low quality

3. In the text, the LD has never been clearly reported, i.e., the indication of 10mg/Kg is only reported in one figure and in the experimental section. Conversely, the indication "30mg/Kg" for the HD was clearly stated also in the text. It is better to report the amount of LD in the text at its first appearance

Author Response

Thank you very much for recommending our work for publication.

Comment: Which is(are) the advantage(s) of 6b with respect to gefitinib that could justify a potential clinical application of the new compound?

Response: As clearly indicated in the results, the effect of the 6b compound was much more significant than that of gefitinib at a similar dose (30mg/Kg). Moreover, the anticancer effect of the 6b compound with a much lower dose (10mg/Kg) was comparable to the 30mg/Kg dose of gefitinib.

The results, therefore, suggest a potentially higher efficacy and clinical application of the 6b compound.

Comment: In the introduction the authors reported a superior activity of 6b in gefitinib-resistant H1975 cells with respect to gefitinib. Nevertheless, they then used a xenograft model of A549 cells which are sensitive to gefitinib. Why? It should have been better and more interesting to evaluate the in vivo potential of 6b in a gefitinib-resistant model. Clearly, I can not suggest to re-run the in vivo experiments on a different cancer model, but this point needs to be properly discussed and clarified in the text

Response: We agree with the reviewer’s comment that it would be interesting to evaluate the in vivo efficacy of 6b in a gefitinib-resistant model.

However, as reported in our previous study (doi.org/10.3390/molecules26051490) highlighting the in vitro anti-proliferative effects of compound 6b, a much better response in terms of IC50 values was observed in A549 cells (IC50 value=2.7 µM) in comparison to that in H1975 (IC50 value=3.65 µM). For this reason, the in vivo efficacy of the compound was first checked and reported in the A549 xenograft nude mice model.

We agree with your valuable comment and mentioned this clarification in the last paragraph of the discussion section of the revised manuscript, indicating the future perspective of the implication of compound 6b in gefitinib-resistant H1975 cells.

Comment: How LD and HD have been selected? Besides, the LD (10mg/Kg) and HD (30 mg/Kg) are not that different. It would have been better to use a ten-fold difference in the two doses. Please clarify

Response: In the case of anticancer compounds, a ten-fold difference between the two doses would result in a highly toxic dose. Therefore, usually, only a two-to three-fold increase in dose is administered.

The HD (30mg/Kg) for compound 6b was chosen as a similar dose to gefitinib (30mg/Kg) in order to compare the potency (which is always done at equivalent doses). The dose of gefitinib (30mg/Kg) was taken from the reported studies (https://onlinelibrary.wiley.com/doi/pdf/10.1002/ijc.11539). Further, to check the efficacy of compound 6b in comparison to gefitinib, a much lower dose (LD=10mg/Kg) was chosen. Interestingly, even the LD group showed a comparable effect to that of gefitinib.

Some minor points:

  1. the English style is not fine, and several sentences need to be rephrased

Response: We have revised our manuscript grammatically and typographically.

  1. check for type errors. Just as an example, in Figure 4B-D there is a mistake (Zefitinib in place of Gefitinib). Also, the figure is of very low quality

Response: We have corrected the mistake. Figure 4 has been revised.

  1. In the text, the LD has never been clearly reported, i.e., the indication of 10mg/Kg is only reported in one figure and in the experimental section. Conversely, the indication "30mg/Kg" for the HD was clearly stated also in the text. It is better to report the amount of LD in the text at its first appearance

Response: We have mentioned LD dose in the text as its first appearance.